



# MESMAR v1: A new regional coupled climate model for downscaling, predictability, and data assimilation studies in the Mediterranean region

Andrea Storto[1,2], Yassmin Hesham Essa[1,3], Vincenzo de Toma[1], Alessandro Anav[2,4], Gianmaria Sannino[2,4],

Rosalia Santoleri[1], Chunxue Yang[1,2]

[1] Institute of Marine Sciences (ISMAR), National Research Council (CNR), Rome, Italy

[2] ICSC – National Research Centre for High Performance Computing, Big Data and Quantum Computing, Italy

[3] Central Laboratory for agricultural climate (CLAC), Agricultural Research Center (ARC), Egypt

[4] Italian National Agency for New Technologies, Energy and Sustainable Economic Development (ENEA), Rome, Italy

*Correspondence to:* Andrea Storto, Institute of Marine Sciences (ISMAR), National Research Council (CNR), via del Fosso del Cavaliere 100, I-00133 Roma, Italy; Email: andrea.storto@cnr.it

**Abstract.** Regional coupled and Earth System models are fundamental numerical tools for climate investigations, downscaling of predictions and projections, process-oriented understanding of regional extreme events, and many more applications. Here we introduce a newly developed coupled regional modeling framework for the Mediterranean region, called MESMAR (Mediterranean Earth System model at ISMAR) version 1, which is composed of the WRF atmospheric model, the NEMO oceanic model, and the HD hydrological discharge model, coupled via the OASIS coupler. The model is implemented at moderate resolution (about 1/12° for the ocean and river routing, while twice coarser for the atmosphere) for long-term investigations. We focus on the evaluation of skill score metrics from several sensitivity experiments devoted to i) understanding the best vertical physics configuration for NEMO; ii) identifying the impact of the interactive river runoff; iii) choosing the best-performing physics-microphysics suite for WRF in the regional coupled system. The modeling system has been developed for downscaling reanalyses and predictions, and for coupled data assimilation experiments. We then formulate and show the performance of the system when weakly coupled data assimilation is embedded in the system (variational assimilation in the ocean and spectral nudging in the atmosphere), in particular for the representation of extreme events like intense mid-latitude cyclones (i.e. medicanes). Finally, we outline plans for future extension of the modeling framework.

**Short Summary.** Regional climate models are a fundamental tool for a very large number of applications and are being increasingly used within climate services, together with other complementary approaches. Here, we introduce a new regional coupled model, intended to be later extended to a full Earth System model, for climate investigations within the Mediterranean

region, coupled data assimilation experiments, and several downscaling exercises (reanalyses, and long-range predictions).





## 1 Introduction

Climate changes are known to pose severe threats to the safety and livelihood of the human population as well as to marine and terrestrial ecosystems. This, in turn, requires increasingly accurate and spatially detailed climate services, many of which require the use of regional climate models (RCMs) capable of achieving horizontal resolutions that global climate models (GCMs) cannot achieve due to computational limits (Giorgi, 1990). RCMs are typically implemented at resolutions of 1-20 km and thus

are generally able to resolve the mesoscale eddies in the ocean and provide a superior representation of the ocean-atmosphere exchanges and local energetics, thus adding value to global climate simulations (e.g., Rockel et al., 2008; Feser et al., 2011; Rummukainen, 2016). RCMs are used to downscale global reanalyses and thus for monitoring purposes (e.g., Rockel, 2015), or to downscale short- and long-term forecasts and climate projections over areas of interest - for example, through the Coordinated Regional Climate Downscaling Exercises (CORDEX) initiatives, see e.g., Ruti et al. (2016); Reale et al. (2022), for the Med-

CORDEX exercise over the Mediterranean region. Indeed, the prediction of high-impact weather and climate events can benefit significantly from the use of RCMs, when, for instance, the enhanced representation of net heat fluxes is important (see e.g., Akhtar et al., 2018). The reader is referred, for example, to the review by Giorgi et al. (2019) for a historical perspective on the development of RCMs and open challenges, such as, uncertainties in high-resolution configurations, misspecification of lateral boundaries and radiative forcing (Foley, 2010), and the use of multimodel simulations (Rummukainen et al., 2015).

In addition, RCMs provide a numerical tool for process-oriented investigations, data assimilation and observing network assessments, and predictability experiments. Coupled data assimilation in regional climate models is still largely unexplored, besides some pioneering applications (see e.g., Li et al., 2020), but represents a high potential for regional predictability gain, linked, among several factors, to the correction of imbalances at initial time and/or at the lateral boundaries, and the maximization of the benefits of the regional observing networks (e.g., Penny et al., 2019).

In the Mediterranean Sea, heat content anomalies are an important precursor of society-impacting and strongly (air-sea) coupled phenomena, such for instance heavy precipitation events (HPEs) and Mediterranean hurricanes (medicanes, e.g., Flaounas et al., 2022). This makes the use of RCMs appealing for both short-range predictability problems and long-term climate applications such as regional reanalyses and the dynamical downscaling of long-range predictions. The retrospective analyses of the most devastating HPEs have outlined the importance of the anomalously warm sea surface temperatures in the Mediterranean

cyclogenesis, which are responsible for enhancing moisture fluxes and convection, associated also to complex interactions with orography (e.g., Lebeaupin et al., 2013; Cassola et al., 2016). The importance of anomalous sea surface temperature (SST) has been proven crucial, especially in the western Mediterranean region, but its impact encompasses all Mediterranean HPEs (e.g., Duffourg and Ducrocq, 2011). Consequently, recent studies have shown that predictive skills of weather forecasts made by numerical weather prediction (NWP) systems significantly increase when the atmospheric models are coupled to ocean models,

allowing interactive feedback with the ocean (Lebeaupin Brossier et al., 2015; Hirons et al., 2018).

Within the Mediterranean region, there is also an important occurrence of severe mesoscale cyclones with tropical-like features, referred to as medicanes (Mediterranean hurricanes) (Flaounas et al., 2022). In a changing climate, such phenomena are expected to increase in intensity (Cavicchia et al., 2014). Coupled models can enhance the correct simulation and prediction of medicanes through e.g., high-resolution downscaling modeling approaches (Cavicchia and von Storch, 2012), correct initialization of

coupled simulations (Ricchi et al., 2017), air-sea feedbacks (Akhtar et al., 2014), including feedbacks between anomalously warm sea surface temperature and features like atmospheric rivers (Flaounas et al., 2022). Additionally, strongly coupled data assimilation (e.g., Storto et al. 2018) is expected to improve the representation of coupled events such as hurricanes, because of the optimization of the observing network (Li and Toumi, 2018; Zhang and Emanuel, 2018) and the importance of the upper ocean heat content in modulating the hurricane intensity (Scoccimarro et al., 2018). Therefore, one focus of the present work is





to evaluate the sensitivity of the prediction of intensity and track of past events of medicanes to different configurations of the
data assimilation system.

In this article, we present the first consolidated version of a regional climate model developed at the Institute of Marine Sciences
(ISMAR) of the National Research Council of Italy (CNR) in collaboration with the Italian National Agency for New
Technologies, Energy and Sustainable Economic Development (ENEA). The system is called MESMAR (Mediterranean Earth
System model at ISMAR) and, in the configuration presented here, includes atmosphere, ocean, and hydrology components at a
spatial resolution of 7-14 km. It covers the Mediterranean basin and is intended for downscaling and predictability exercises, and
as a testbed for coupled data assimilation experiments. In the following sections, we detail the configuration of the system
(section 2), and the results from a few notable sensitivity experiments that led to the reference configuration (section 3); we then
assess the ocean heat budget in the reference, assimilation-blind, experiment (section 4). Next, we focus on the configuration and
assessment metrics in a series of weakly coupled assimilation experiments (section 5). Finally, Section 6 concludes and discusses
the main achievements and plans.

## 2 Earth system model configuration

We detail in this section the configuration of the coupled model components, including the coupler settings.

### 2.1 Atmospheric model

The atmospheric model component is the Weather Research and Forecasting (WRF) community model, version 4.3.3
(Skamarock et al., 2021), implemented over the Mediterranean and European regions at 15 km of horizontal resolution and 41
vertical hybrid levels. The WRF domain (Figure 1) extends from Northern Africa (south) to the middle of the Scandinavian
peninsula (north), and from the North Atlantic (west) to western Asia (east). The domain is adopted after Anav et al. (2021),
using the standard stationary geophysical fields provided by WRF through the WRF Preprocessing System (WPS) package.
The WRF timestep is set equal to 60 s. The suite of physical, microphysical, and subgrid parametrization options comes in most
cases from the MED-CORDEX sensitivity experiments (see e.g. Fita et al., 2019, for an older setup) or has been specifically
tested (see also section 3). In particular, the Thompson et al. (2008) microphysics is used, while the radiation is modeled with the
RRTMG Rapid Radiative Transfer Model (Iacono et al., 2008). The land model component is the Noah-MP (multi-physics) Land
Surface Model (Niu et al., 2011) with four soil layers. The Mellor-Yamada turbulent closure of Nakanishi and Niino (2006) is
adopted, while the Grell and Freitas (2014) cumulus parametrization is used.

Lateral boundary conditions are imposed over the ten grid-points closer to the four boundaries. In hindcast mode (e.g., for
simulations or reanalysis downscaling), the lateral boundary conditions are taken from 3-hourly fields of the ECMWF ERA5
reanalysis (Hersbach et al., 2020).

### 2.2 Ocean model

The primitive equation NEMO ocean model (Madec et al., 2017), version 4.0.7, is the ocean model component of MESMARv1,
developed and maintained by the homonymous NEMO consortium. The model covers the Mediterranean Sea region from an
Atlantic box to the Dardanelles (see Figure 1) over a regular domain at a horizontal resolution of about 7 km and with 72 vertical
depth levels with partial steps. The (baroclinic) model timestep is set to 450 s, while the barotropic timestep is equal to 6 s, using
the split-explicit free surface scheme (Shchepetkin and McWilliams, 2005).



In our setup of NEMO, the shortwave radiation extinction coefficients are specified using the 3-band spectral discretization of Morel and Maritorena (2001), with coefficients that depend on the chlorophyll concentration, taken in turn from the level-4 (L4) monthly fields of Brewin et al. (2015) and distributed by the Copernicus Marine Service. Horizontal diffusivity (modeled with a Laplacian operator) and viscosity (modeled with a bi-Laplacian operator) coefficients are set equal to 80 m s$^{-2}$ and 4.5E+9 m$^2$ s$^{-4}$, respectively. These values are increased by 20 % in the proximity of the Gibraltar Strait and the Aegean Sea. The Generic

Length Scale (GLS) scheme (Umlauf and Burchard, 2003) is used for vertical mixing; GLS is a general framework for vertical mixing, and we adopt the Mellor and Yamada (1982) turbulence closure, with the stability function of Canuto et al. (2001).

In hindcast mode, the lateral boundary conditions are provided by the ECMWF ocean reanalysis ORAS5 (Zuo et al., 2019). Lateral boundary conditions are imposed as follows: barotropic velocities through the Flather scheme (Flather, 1994); baroclinic velocities specified at the boundary gridpoints from the external sources; temperature and salinity through a flux relaxation

scheme that gradually relax the tracer fields towards the external fields over the ten inner gridpoints closer to the boundaries. ORAS5 was chosen among the Copernicus Marine Service GREP (Global Reanalysis Ensemble Product, Storto et al., 2019) as it provides the best sea surface height validation skill score statistics against altimetry data compared to the other reanalyses. In preliminary experiments, the average root-mean-square error of 3.5 cm was found when using ORAS5, compared to 4.0 cm when other reanalyses were used (not shown), while temperature and salinity have comparable skill scores across the

experiments using different GREP reanalyses as lateral forcing.

The river runoff is imposed through the Hydrological Discharge model (see next section), except at the Dardanelles, where it is set equal to the monthly climatology of the Black Sea outflow into the Mediterranean Sea, as given by Kourafalou and Barbopoulos (2003).

**2.3 Hydrological discharge model**

MESMARv1 includes interactive river runoff, estimated by the Hydrological Discharge (HD) model, version 5.1 (Hagemann et al., 2020), developed and maintained by Helmholtz-Zentrum Hereon. The HD model implements a horizontal resolution of 1/12° degree over the European continent and it contains specific developments for coupled simulations, including the support for the OASIS coupler (Ho-Hagemann et al., 2020). Its hydrological core stems from the MPI model (Hagemann and Dümenil Gates, 2001).

The HD timestep is set to 30 minutes, which is generally a higher frequency than most implementations, and it is chosen to ease the coupler exchanges (see below). A discharge-dependent river flow velocity is used. Additionally, we have modified the routine responsible to map the river discharge onto oceanic NEMO points to include a smoothing function, which conservatively spread the discharge from one ocean point to the 25 neighboring grid-points. This is required to avoid instability problems where large discharge occurs.

**2.4 Coupler**

The coupler used by MESMARv1 is OASIS (OASIS3-MCT_5.0, Craig et al., 2017), which is a flexible parallel coupler developed by CERFACS. We use first-order conservative remapping to interpolate all the fields exchanged from one model to the other; the coupling frequency is set to 30 minutes for all fields.

Table 1 summarizes the fields exchanged through the coupler from and to the different model components. Note that the air-sea

fluxes over the oceans are computed within WRF, following the surface scheme of Janjić (1994). Additionally, the skin sea surface temperature scheme of Zeng and Beljaars (2005) is adopted to diagnose the diurnally varying skin SST within the WRF





bulk formulas. WRF passes also the atmospheric pressure fields to NEMO, for the latter to account for the inverse barometer effect in the sea level computations (see e.g., Wunsch and Stammer, 1997). Note also that WRF communicates to HD to provide fields of surface and sub-surface runoff, which are then remapped onto the ocean grid and passed to NEMO after the river
routing scheme is run in HD.

### 2.5 Experimental setup

Depending on the specific application and test, several setups have been used for MESMARv1. In general and unless otherwise specified, atmospheric initial conditions are provided by the ERA5 atmospheric reanalysis; oceanic initial conditions from the GLORYS12 ocean reanalysis (Lellouche et al., 2021); river routing initial conditions from a previous standalone run of the HD
model (that is, run in uncoupled mode with input surface and subsurface runoff from a standalone WRF simulation); coupler initial conditions (i.e., the exchanged fields) are set to zero. Lateral boundary conditions are given by ERA5 and ORAS5, for the atmospheric and oceanic lateral forcing, respectively, as specified in Sections 2.2 and 2.3, unless otherwise specified.

### 3 Sensitivity experiments

Selected sensitivity experiments are presented in this section, to provide a rationale for the choice of individual schemes or
parametrizations.

### 3.1 Impact of the interactive river discharge

The effect of the interactive river runoff is summarized in this section. In particular, we have tested for 2 years (2015-2016) the use of the climatological runoff, taken from the ORCA12 standard configuration of NEMO and adapted by Bourdalle-Badie and Treguier (2006) from the Dai and Trenberth (2002) compilation of river runoff data. This experiment corresponds to the
uncoupled runoff, namely what is customarily done in most oceanic applications (e.g. ocean reanalyses, see e.g. Storto et al., 2019), and it is compared with the standard MESMAR configuration where the river discharge is provided interactively by the HD model. This exercise aims at assessing the qualitative impact of the WRF-HD-derived runoff; however, assessing the impact of the inter-annual variations requires dedicated multi-decadal experiments, which are expensive and beyond the scope of the present general manuscript. Thus, we mostly verify that the land-ocean coupling configuration leads to satisfactory results in
terms of the Mediterranean freshwater budget.

Differences between the climatological river runoff and the one derived from WRF-HD are visible in Figure 2, in terms of total discharge and area-averaged sea surface salinity for the whole Mediterranean Sea (excluding the Atlantic box from the model domain). In general, the interactive land-ocean coupling leads to a total discharge smaller than that with climatological runoff, and a shift of the minima/maxima of the yearly cycle (minima from September-October to November-December, maxima from
February-May to April-August). Compared to the bias-corrected river discharge from JRA55-do (Tsujino et al., 2020), the difference in discharge is lower than the ORCA12 climatology, at least for the year 2016, but the seasonal offset is more pronounced. Accordingly, the sea surface salinity (SSS) increases year-round and results in a lower bias compared to the UKMO EN4 SSS (Good et al., 2013), as shown in Figure 2 (bottom panel). In particular, the time-averaged map of sea surface salinity anomalies (Figure 3) caused by the interactive land-ocean coupling highlights the salinity increase in several coastal areas of the
Mediterranean Sea, particularly the Gulf of Lyon, the Adriatic Sea, and the Levantine basin, with values exceeding 1 psu along the major river mouths of the Mediterranean basin. Slight freshening of the surface waters is visible in the Aegean Sea, off





Sicily, and in front of major lagoons (Akyatan and Lake of Tunis), but the overall effect of the interactive river discharge is a salinification of the surface waters by 0.06 psu on average, during the 2-year study period.

Skill score metrics computed against all available in-situ profiles extracted from the UKMO EN4 profile dataset are shown in
Figure 4. Profiles of bias and RMSE of salinity confirm the positive impact of the land-ocean coupling that penetrates up to about 200 m of depth. Fresh biases are significantly mitigated in the top 100 m, while RMSE shows improvements from the surface to the halocline. The results indicate that the WRF-HD-NEMO system has great potential for improving the representation of the water cycle in the Mediterranean region. Indeed, the use of the interactive river runoff allows us to close the water cycle in the regional basin, besides the improvement of the performances of the regional coupled model in representing the
salinity variations.

### 3.2 NEMO vertical physics

Several sensitivity tests were performed to identify the best vertical mixing configuration for the NEMO model. Here, we show the results from the two best-performing implementations of the GLS scheme (as described in section 2.2), and the TKE scheme, implemented with the same background coefficients as in Storkey et al. (2018). Figure 5 shows the winter and summer bias of
sea surface temperature computed against the daily SST analyses from the Copernicus Marine Service (Pisano et al., 2020). Compared to the GLS scheme, the TKE induces enhanced mixing in summer (that is, weakened stratification), with colder biases in most areas and notably in the Adriatic Sea. GLS has overall positive and smaller biases than TKE in the southern part of the domain, which leads to stronger stability of the model (that is, stronger stratification). In winter, TKE has a relevant warm bias, especially in the western basin. These biases are propagated onto the near-surface air temperature (not shown), indicating that the
GLS vertical mixing implementation has a better impact over the sea, while the differences led by the use of the two schemes are in general negligible over land, where biases are dominated by other factors, such as land surface processes (e.g. Davin et al., 2016).

The surface signature is confirmed by the skill scores profiles against in-situ data (Figure 6), which highlights the smaller biases obtained with GLS than TKE, up to about 800 m of depth. Looking at salinity, the TKE shows too many salty waters year-round.
Furthermore, the RMSE is smaller with GLS in the top 50 m of depth and between 150 and 600 m of depth. At the sea surface, both simulations are too salty compared to the mean observed profile in the upper 50 m (top middle panel of Figure 6), yet the GLS scheme significantly mitigates the salinity over-estimation; elsewhere, the impact is neutral. Year-round, the improvements in temperature are visible to about 200 m of depth. The TKE-enhanced mixing leads to less sharp thermocline compared to GLS and the mean observed profile. Moreover, GLS shows a significant bias reduction (0.4°C with GLS against almost 1°C with
TKE) on the temperature peak in the upper 50m.

### 3.3 WRF configuration

In the initial phase of the MESMAR implementation, we performed many sensitivity experiments, both coupled and uncoupled, to identify the best-performing suite of physics and microphysics schemes. Here, we report results from the configurations for which we performed long experiments (1993-2021) in coupled configuration (i.e., with NEMO and HD). Further to the
configuration described in detail in Section 2 (and called REF), we performed two other experiments: the first (W01) has different microphysics (Morrison 2 moments), surface layer (Revised MM5 scheme), boundary layer (YSU) and cumulus (Betts-Miller-Janjic) schemes, compiled together similarly to a previous configuration of WRF as in Anav et al. (2021). The second experiment (W02) is as W01, but with the less advanced NOAH land surface model, replacing NOAH-MP in W01 and REF.





Table 2 and Figures 7-9 report validation statistics and climatology maps, for wind speed at 10 m, air temperature at 2 m, and
total precipitation, compared to the E-OBS terrestrial dataset (Cornes et al., 2018).

The comparison with E-OBS wind speed (Figure 7) indicates, year-round, that all experiments have a positive bias and that the
best performances are achieved by REF, which shows a rather low bias between 0 and 2 m s-1 (0.74 m s-1 on average, see Table
2). The other two experiments (especially W02 in autumn and wintertime) exhibit more pronounced positive biases. Differences
in 2m temperature performances are smaller (Figure 8), with W02 on average outperforming the two others. All the experiments
reproduce the winter cold bias on North-Eastern Europe, already found in several configurations of WRF (see e.g. Anav et al.,
2021), with W01 exhibiting the coldest bias therein. Unlike W01 and W02, REF does not exhibit a large warm bias in
summertime over Europe. Finally, in the comparison with precipitation from E-OBS (Figure 9), REF is found to be the wettest
model, especially in the summer. Although there is no configuration better than the others concerning all atmospheric
parameters, the reference configuration is chosen, as it provides the best near-surface atmospheric circulation and keeps
reasonably low biases in air temperature.

## 4 Reference Simulation

In this Section, we evaluate the Mediterranean Sea warming and heat budget for the period 1993-2021 from a long MESMAR
simulation, which implements the optimal configuration of WRF and NEMO as detailed in the previous section. The
Mediterranean Sea is a climate change hot spot, which warms at a higher rate than the global ocean (Lionello and Scarascia,
2018), and whose warming is expected to accelerate in the future (Cos et al., 2022). Thus, assessing the potential of the coupled
regional model in capturing the ocean heat content (OHC) variability is a fundamental exercise to validate its applicability for
climate monitoring.

The reference simulation does not contain any observational constraint, besides the lateral boundaries forced to the ECMWF
ERA5 and ORAS5 atmospheric and oceanic reanalyses. Therefore, we do not expect that the warming rate is close to that
observed, as both the atmosphere and ocean models are free to evolve following their internal physics; however, the coupled
model simulation may still be able to capture to some extent the interannual variations of the ocean heat content.

The top left panel of Figure 10 shows the OHC from MESMAR and, for comparison, the OHC compiled as Ocean Monitor
Indicator (OMI) of the Copernicus Marine Service (https://doi.org/10.48670/moi-00261). The OMI is the ensemble mean of
several global and regional sources that include both objective analyses and reanalyses. Results show that the increase of OHC is
underestimated in MESMAR: the warming rate for the full period, calculated as the linear trend of OHC, is 1.39 W m-2 in the
OMI and 0.24 W m-2 in MESMAR. However, the interannual variations of OHC match very well between the two timeseries.
This is shown by the dashed red curve in the top left panel of Figure 10, which is the MESMAR interannual variations with the
linear trend rectified to match that of the CMEMS OMI. In this case, the inter-annual variations almost overlap with those from
CMEMS OMI. Events, like the 2002-2005 cooling and the successive sharp warming during 2006-2011 (mostly due to the North
Atlantic forcing variability, see e.g. Iona et al., 2018) are, indeed, well captured.

To understand the representation of the causes of the warming in MESMAR, we have analyzed separately the two warming
sources in the Mediterranean basin, using a box approach where OHC tendencies equal the sum of lateral heat transports and net
downward air-sea heat flux, and assuming that heat contributions from rivers and Dardanelles Strait is negligible (Harzallah et
al., 2018). The bottom panel of Figure 10 shows the net downward air-sea heat flux in MESMAR and ERA5; the interannual
variations in the two datasets are very well correlated; however long-term values indicate an important under-estimation of the
MESMAR net fluxes, equal to -5.04 +/- 4.99 W m-2 against 4.15 +/- 4.78 W m-2 (ERA5), namely the average difference is large



and exceeds 9 W m$^{-2}$. It also should be noted that during the first five years, the net heat flux in ERA5 appears unrealistically large, even exceeding 10 W m$^{-2}$. It is well known that the ensemble dispersion of models and reanalyses in simulating the net heat flux is very large (e.g., Harzallah et al., 2018); however, long-term closed heat budget in the Mediterranean Sea requires the

net heat flux to be slightly negative in order to compensate the positive heat inflow from Gibraltar Strait (see Jordà et al., 2017, and later in this section), implying an over-estimation of ERA5 and an under-estimation of MESMAR.

In terms of heat transport, the top right panel of Figure 10 shows the incoming heat transports at Gibraltar Strait, from MESMAR and the Copernicus Marine Service regional reanalysis (Escudier et al., 2021). The two timeseries show close variations and equal long-term means (between the error bars), equal to 5.34 +/- 0.44 W m$^{-2}$ (MESMAR) and 4.97 +/- 0.43 W m$^{-2}$ (CMEMS).

The values are also well aligned with other in-situ and model-based estimates, as for instance 5.2, 5.0, and 4.9 (respectively: MacDonald et al., 1994; Astraldi et al., 1999; Harzallah et al., 2018).

The very close values of lateral incoming heat transports mean that differences are due only to the atmospheric radiative forcing into the ocean. In particular, MESMAR leads to too small air-sea flux, while the use of ERA5 to much too warm flux. The CMEMS reanalysis instead, which assimilates data, can rectify the fluxes.

Concerning regional warming (Figure 11, left panels) MESMAR provides similar patterns compared to the CMEMS reanalysis, with maximum warming on the eastern side of the Mediterranean. This confirms the ability of the coupled regional model to capture inter-annual variations and spatial patterns. The two right panels show the net air-sea heat flux, which reveals that the MESMAR underestimation of heat uptake from the atmosphere is rather homogenous, as patterns are close to those of the ERA5 reanalysis. In particular, we found that both the turbulent fluxes (sensible and latent heat) are, together, overestimated by about 5

W m-2 in MESMAR, and the incoming solar radiation is underestimated by another 5 W m-2, compared to ERA5.

## 5 Data assimilation

### 5.1 Weakly coupled assimilation configuration

One important application of regional coupled models is the possibility to downscale multi-decadal climate reconstructions from both atmospheric and oceanic reanalyses (e.g. Vannucchi et al., 2021) and short-range predictability studies. To this end,

MESMAR implements a weakly coupled data assimilation system, where the oceanic state is constrained by a three-dimensional variational (3DVAR) data assimilation system (Storto et al., 2018), and the atmospheric state by a spectral nudging scheme (Choi and Lee, 2016), which is already part of the WRF modeling system.

The 3DVAR scheme implements stationary background-error covariances estimated from the dataset of differences between two long-term simulations with different physics options in both the WRF and NEMO configurations. In particular, this anomaly

dataset is obtained for the period 1994-2020 from the differences between two experiments with different ocean and atmospheric physics, shown in the previous sections (one configuration embedding the TKE vertical mixing scheme and other atmospheric schemes as in W02, see Section 3.3). Preliminary tests (not shown) indicated that using pairs of experiments with different physics for estimating background-error covariances led to better skill scores than the use of climatological anomaly differences from a long-term simulation (see Storto et al., 2014, for a discussion on the approach to estimate background-error covariances).

Background-error covariances are modeled through the application of multi-variate spatially-varying EOFs - for vertical covariances -, and first-order recursive filter with spatially-varying correlation length-scales for the horizontal correlations, as in Storto et al. (2014).





The assimilated observations include all in-situ profiles (XBT and CTD casts, moorings, floats, and gliders), extracted from the UKMO EN4 dataset (Good et al., 2013). Observational errors and variational quality control are adopted as in Storto (2016),

which allows for non-linear weighting of the observations.

At the sea surface, a relaxation scheme is applied to correct air-sea heat and freshwater fluxes by nudging the sea surface temperature (SST) and sea surface salinity (SSS) to SST and SSS analyses, taken from the CNR ISMAR SST analyses (Pisano et al., 2020) and the UKMO EN4 objective analyses (Good et al., 2016), respectively. The relaxation time scales are set equal to 15 and 300 days for SST and SSS, respectively, after several preliminary sensitivity experiments aimed to identify the best-scoring

configuration (not shown).

In the atmosphere, a spectral nudging scheme is applied in WRF, which nudges the large-scale component of wind, temperature, and humidity toward the ECMWF ERA-5 reanalysis (Hersbach et al., 2020). Large scales are defined based on fast Fourier transform (FFT) decomposition, with the 6 and 5 wavenumber cutoffs, which are equivalent to about 850 km in MESMAR (see e.g. Omrani et al., 2015, for more information on the WRF spectral nudging capability). The nudging time scale is equal to 1

hour for wind and temperature, and 1 day for humidity. For comparison, full-field nudging is also shown in the next section, to evaluate different ways to constrain the atmospheric fields.

The assimilation time window is set to 3 days, namely every three days the ocean state is corrected employing the 3DVAR analysis increments; in reanalysis mode, the atmospheric spectral nudging is continuous and uses three-hourly fields from ERA5.

## 5.2 Experiments and results

Several experiments have been performed to identify the best-scoring configuration for oceanic and atmospheric data assimilation. Here, we show only the impact of activating different components of data assimilation, combining, in particular, the cases of no-assimilation (OC0) and assimilation (OC1), in the ocean, and no-assimilation (AT0), full-field (AT1) and spectral nudging (AT2), in the atmosphere. The summary of experiments, along with selected validation skill scores, is reported in Table 3. All these experiments have been run for 3 years (2018-2020) and initialized from the reference simulation shown in Section 4.

Skill scores in Table 3 indicate slight improvements (1% to 2%) on the atmospheric skill scores when the ocean data assimilation is switched on (AT0OC1 versus CTRL), and similarly for the impact on oceanic skill scores when atmospheric data assimilation is active (e.g. AT2OC0 versus CTRL). The largest impact on the skill scores is achieved when the assimilation of each model component is active. It is worth noting that spectral nudging leads to slightly worse accuracy for temperature and wind, while RMSE remains unchanged for the geopotential. This is implicit in the scale selective constraint of the spectral nudging; however,

SST skill scores are most benefited by the spectral nudging, suggesting that full-field nudging may, to some extent, interfere negatively with the air-sea flux computation.

Figure 12 details the bias and RMSE profiles in the atmosphere for the six experiments, verified against radiosonde data from the NOAA/ESRL Radiosonde Database (https://ruc.noaa.gov/raobs/). Spectral nudging provides less biased near-surface air temperature values, although RMSE is the smallest with full-field nudging, indicating that the temporal variability is better

captured in the latter case. Qualitatively similar results hold for wind speed, while humidity skill scores are less impacted by the data assimilation settings, partly due to the smaller nudging coefficient than for other parameters, and the dominating effect of microphysics parameterizations.

Seawater temperature and salinity skill scores are presented in Figure 13, as profiles of mean state, bias, and RMSE. Salinity is characterized by salty biases in all assimilation-blind experiments in the top 100 m of depth. The assimilation scheme

successfully corrects the bias and approximately halves the RMSE in the upper ocean. For temperature, the CTRL experiment is characterized by sea surface cold bias, while experiments with only atmospheric data assimilation are characterized by warm





bias. The adoption of variational ocean data assimilation rectifies both types of bias and leads to consistently small RMSE throughout the water column. The benefits of the different assimilation schemes on the SST skill scores are shown in Figure 14 for two selected pairs of experiments as RMSE differences. The RMSE is calculated against SST analyses from satellite data
(Pisano et al., 2020). The top panel shows the impact of spectral versus full-field nudging (positive values indicate the superiority of spectral nudging). In most areas of the Mediterranean Sea, and dominantly in the western part of it, spectral nudging outperforms full-field nudging, likely due to the effective spatial resolution which is not degraded in the full-field nudging. The impact of ocean data assimilation (bottom panel) is large and rather homogenous throughout the model domain, peaking east of Gibraltar Strait.

To better understand how spectral nudging is not disruptive to the upper ocean circulation, Figure 15 shows the eddy kinetic energy (EKE) from the different experiments, calculated from the sea surface height using the geostrophic velocities (e.g. Wang et al., 2019). The timeseries show that ocean data assimilation significantly impacts the EKE, although altimetry is not assimilated, as in previous global ocean studies (Storto et al., 2016). Such an increase is in the range of 44-48% depending on the atmospheric data assimilation configuration. However, while full-field nudging leads to a decrease of EKE of about 2.5%
(AT1OC1 versus AT0OC1), spectral nudging provides an additional 3% increase (AT2OC1 versus AT0C1), indicating its slight benefits in reproducing the mesoscale ocean circulation. This is also confirmed by the validation against surface current speed from drifters (not shown), which highlights a slight improvement (of the order of 1-1.5%) when spectral nudging and ocean data assimilation are adopted, compared to the CTRL or full-field nudging experiments.

### 5.3 Impact on the representation of Mediterranean hurricanes

We conclude our assessment with the skill scores relative to the representation of Mediterranean hurricane (medicane) events. In particular, during the 2018-2020 period, two events of strong intensity occurred in the eastern part of the Mediterranean. These two events are the Zorbas (27 September-2 October 2018) and the Ianos (14-21 September 2020) medicanes. In particular, we looked at the reanalysed and forecasted events, also in comparison with the ECMWF ERA5 reanalysis, for the different assimilation configurations presented earlier.

The top panels of Figure 16 show the two medicanes' tracks – calculated as the location of the minimum sea level pressure – from the observed best track and the experiments with atmospheric data assimilation, in reanalysis mode (i.e., continuous data assimilation). CTRL and AT0OC1 are not shown as their error in reproducing the medicane tracks are very large. Table 4 summarizes medicane verification skill scores. All experiments can capture the tracks of the medicanes, with positioning errors of the order of 36-38 km and 23-31 km for the two events, respectively. The smallest distance errors are for the AT2OC0 and
AT1OC1, respectively, although the differences are small. However, spectral nudging provides the best skill scores for the minimum pressure and the maximum wind speed, visible in Table 4 and the bottom panels of Figure 16. Additionally, ocean data assimilation further improves the representation of the baric minima for both events, leading to another 10% improvement in terms of pressure minima RMSE. To a lesser extent, the improvement occurs also for wind speed maxima (about 3% improvement). These results indicate that while the adoption of atmospheric spectral nudging is crucial in capturing the
medicane evolution, namely its track, ocean data assimilation can provide a significant additional improvement in capturing the intensity of the events. This proves the added value of the coupled modeling and the potential of coupled data assimilation to increase medicane predictability.

Similar diagnostics have been assessed and calculated in forecasting mode for the Zorbas medicane. In particular, several forecasts were initialized on 28 September from the initial conditions provided by their respective assimilation experiments, and
the unconstrained coupled model without any data constraint was then run in forecasting mode for the following 5 days. Results





are summarized in Figure 17 in terms of forecasted track and RMSE decreases compared to the corresponding ERA5 forecasts. The spectral nudging can better capture the medicane landing, while full-field nudging significantly deviates the track southwards. The use of ocean data assimilation provides a small impact on the forecasted track; however, in terms of mean sea level pressure and wind speed forecasts, there occurs significant improvement when the initialization includes oceanic observations, of about 10% RMSE decrease (compared to ERA5) for both parameters. This confirms the non-negligible potential of oceanic data assimilation on hurricane predictability (Zhang and Emanuel, 2018).

## 6 Summary, discussion, and future extensions

In this work, we have introduced the configuration of a new high-resolution regional climate model for the Mediterranean region (MESMAR) and presented several assessment results. While there exist already several regional coupled and climate models over this region (e.g., Lionello et al., 2003; Lebeaupin and Drobinski, 2009; Artale et al., 2010; Akhtar et al., 2018; Nabat et al., 2020; Reale et al., 2020; Anav et al., 2021), our goal is to set up an affordable numerical framework, to be possibly upgraded in the future, to study the predictability of specific events through downscaling exercises and state-of-the-art coupled data assimilation algorithms. The main objective of the present work is to present the configuration and the basic performances of the system and evaluate weakly coupled data assimilation experiments.

The model is composed of WRF, NEMO, and HD as atmospheric, oceanic, and hydrology components, respectively, implemented at 15 km, and $1/12°$ of horizontal resolution. Several sensitivity experiments have been performed to identify the optimal coupled model configuration. Our non-exhaustive selection focused on the benefits of a re-tuned oceanic vertical mixing scheme, the positive impact of interactive river discharge on the upper ocean salinity skill scores, and the physics-microphysics parametrizations' suite of WRF on near-surface biases. We have shown that with the optimal configuration, the spatial and temporal variability of the ocean heat uptake is well captured for the period 1993-2021, although some offset in the air-sea net heat fluxes exists, providing an ocean warming weaker than observed in the regional climate model.

Next, we have implemented and assessed a weakly coupled data assimilation system, where atmospheric data assimilation is formulated in terms of scale-selective (spectral) nudging to relax WRF towards the ECMWF ERA5 reanalyses at the scales of about 850 km and larger. The oceanic data assimilation component includes a variational scheme capable to assimilate all observations available in the Mediterranean Sea, with a temporal frequency and assimilation window of 3 days. In a series of 3-year experiments combining different setups of the atmospheric and oceanic data assimilation, we have demonstrated the benefits of the spectral nudging on sea surface skill scores, oceanic eddy kinetic energy, and medicane event representation, while the ocean data assimilation is found crucial not only in the oceanic skill score metrics but also for medicane intensity predictions and, to some extent, to the low-troposphere skill scores. The final configuration including spectral nudging and ocean variational data assimilation will serve as the basis for regionally downscaling global atmospheric and oceanic reanalyses from ECMWF, and as the basis for downscaling monthly to seasonal predictions.

Future extensions of MESMAR will go mostly in three directions. First, the horizontal resolution of the models: while our long-term applications make it difficult to reach a convection-resolving spatial resolution, the atmospheric model resolution could be increased to reduce the spatial resolution factor compared to NEMO. Second, the regional climate model can be extended to include other model components and to turn into an Earth System model (ESM); for instance, wave modeling components and biogeochemical modeling can be embedded in the system to provide an ESM correspondence of MESMAR. Finally, the system is being upgraded to include a strongly coupled data assimilation system, where the data assimilation state vector and the observation operators seamlessly include both atmospheric and oceanic parameters (as in Storto et al., 2018). This will pave the





way for a systematic assessment of the impact of coupled observation operators and initial conditions in both short and long-

range prediction systems and will require preliminary studies on the optimal characterization of the coupled background-error

covariances.





*Acknowledgments.*

This work has received funding from the POR Lazio Program, through the grant A0375-2020-36508 (project "DYNAMOL": observation-aware DYNAMical downscaling of seasonal predictions Over the Lazio area) and from ICSC – Centro Nazionale di Ricerca in High Performance Computing, Big Data and Quantum Computing, funded by European Union – NextGenerationEU. Acknowledgment is made for the use of ECMWF's computing and archive facilities in this research.

*Code availability*

The NEMO ocean model code (v4.0.7) is available at https://forge.ipsl.jussieu.fr/nemo/wiki , the WRF atmospheric model code (v4.3.3) is available at https://github.com/wrf-model/WRF , the HD hydrological discharge model (v5.1) is available at https://zenodo.org/record/5707587#.Y-0VQ3bMKUk .

The frozen version of the MESMARv1 code used in this manuscript is available at: https://doi.org/10.5281/zenodo.7898938 .

Data and scripts used within the manuscript are available at: https://doi.org/10.5281/zenodo.7899115 .

*The authors declare no competing interests.*

*Author contribution*

AS, YH, VdT, CY and AA have contributed to the model developments and model validation; GS and RS have provided guidance on the model developments. AS drafted the initial version of the manuscript, all coauthors have revised the manuscript.





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





| Field | From | To | Notes |
|---|---|---|---|
| Sea surface temperature | NEMO | WRF | Bulk temperature in NEMO at first ocean model level |
| Surface zonal current | NEMO | WRF | For use in the air-sea flux computations that consider the relative wind |
| Surface meridional current | NEMO | WRF | For use in the air-sea flux computations that consider the relative wind |
| Wind stress modulo | WRF | NEMO | - |
| Zonal wind stress | WRF | NEMO | - |
| Meridional wind stress | WRF | NEMO | - |
| Freshwater flux | WRF | NEMO | Given as evaporation minus precipitation |
| Solar heat flux | WRF | NEMO | Penetrative component of the air-sea heat flux |
| Non-solar heat flux | WRF | NEMO | Non-penetrative component of the air-sea heat flux |
| Atmospheric surface pressure | WRF | NEMO | For use to model the inverted barometer effect in NEMO |
| Surface runoff | WRF | HD | From the NOAH-MP land model |
| Subsurface runoff | WRF | HD | From the NOAH-MP land model |
| Runoff at the river mouth | HD | NEMO | Remapped and spread over the NEMO gridpoints |

**Table 1. Fields exchanged through the OASIS coupler between the different model components.**



| Exp | Schemes | Wind speed | 2m Temperature | Precipitation |
|---|---|---|---|---|
| W01 | Microphysics: Morrison (2 moments) Surface layer: Revised MM5 Monin-Obukhov scheme Boundary layer Scheme: YSU scheme Cumulus scheme: Betts-Miller-Janjic scheme | 0.99 (1.07, 0.93) | 1.27 (1.92, 1.41) | 0.36 (0.43, 0.37) |
| W02 | As W01, but with the NOAH land sea model instead of NOAH-MP | 1.57 (1.93, 1.30) | 1.11 (1.38, 1.13) | 0.43 (0.47, 0.48) |
| REF | As described in the text | 0.74 (0.78, 0.69) | 1.23 (1.88, 0.95) | 0.58 (0.40. 1.03) |


**Table 2. List of sensitivity experiments performed, with the list of physics and microphysics parametrizations used in WRF and mean absolute error results against the E-OBS dataset, for wind speed (m s$^{-1}$), 2m temperature (K), and precipitation (mm day$^{-1}$). MAE values report the statistics year-round and, into brackets, for winter (DJF) and summer (JJA) separately.**






| Experiment Name | Atmospheric Assimilation | Oceanic Assimilation | RMSE | | | | | |
|---|---|---|---|---|---|---|---|---|
| | | | T850 | WS 1000-850 | Z500 | SST | T0-50 | S0-50 |
| CTRL | No | No | 2.07 | 3.58 | 29.5 | 0.63 | 1.13 | 0.32 |
| AT0OC1 | No | 3DVAR+SRF | 2.04 | 3.58 | 29.0 | 0.27 | 0.83 | 0.20 |
| AT1OC0 | Full-field nudging | No | 0.82 | 2.08 | 10.7 | 0.71 | 1.10 | 0.32 |
| AT1OC1 | Full-field nudging | 3DVAR+SRF | 0.83 | 2.08 | 10.7 | 0.29 | 0.76 | 0.20 |
| AT2OC0 | Spectral nudging | No | 1.0 | 2.77 | 10.7 | 0.63 | 1.11 | 0.29 |
| AT2OC1 | Spectral nudging | 3DVAR+SRF | 1.0 | 2.77 | 10.7 | 0.27 | 0.80 | 0.20 |

**Table 3. List of experiments performed and shown in section 5 of the text, with different assimilation setups (AT0, AT1, AT2 refer to no atmospheric data assimilation, full-filed nudging, and spectral nudging, respectively; OC0 and OC1 to no oceanic data assimilation and variational ocean data assimilation, respectively). Right-side columns report total skill scores as RMSE for some selected parameters: air temperature at 850 hPa (K), wind speed in the layer 1000-850 hPa (m s⁻¹), 500 hPa geopotential (m), SST (°C), seawater temperature (°C) and salinity (psu) in the top 50 m of depth.**






| Experiment Name | Zorbas medicane | | | Ianos medicane | | |
|---|---|---|---|---|---|---|
| | Position | Pressure | WindSpeed | Position | Pressure | WindSpeed |
| AT1OC0 | 38.12 | 6.1 | 6.6 | 24.0 | 8.7 | 10.7 |
| AT1OC1 | 37.28 | 6.1 | 6.5 | 23.5 | 8.6 | 10.6 |
| AT2OC0 | 36.37 | 3.7 | 3.8 | 24.1 | 6.7 | 8.2 |
| AT2OC1 | 36.47 | 3.2 | 3.7 | 27.7 | 6.1 | 7.9 |
| ERA5 | 36.43 | 6.2 | 7.4 | 31.1 | 8.8 | 12.0 |

**Table 4. RMSE values calculated for the different data assimilation experiments presented in the text in reanalysis mode (i.e., with continuous data assimilation) and the ECMWF ERA5 reanalysis, for the two medicane Zorbas (September-October 2018) and Ianos**
**(September 2020). Parameters assessed are the position (distance with the best-observed track, in km), along-track sea level pressure (hPa), and near-track maximum wind speed (m s$^{-1}$).**

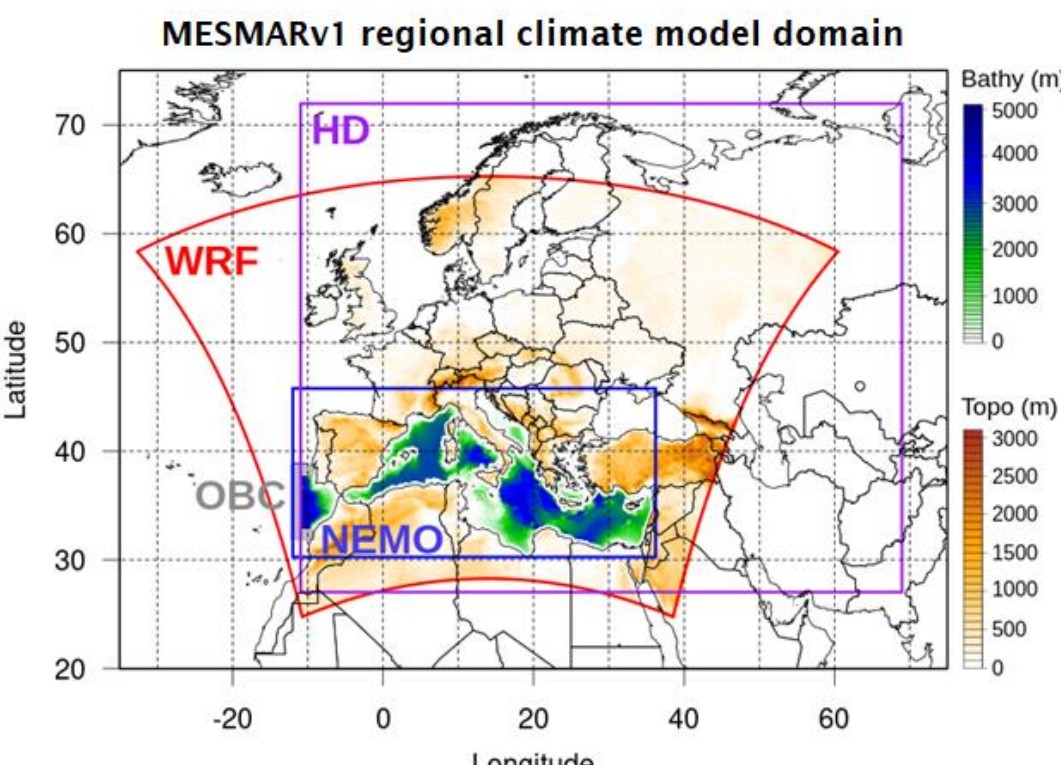


**Figure 1. Computational domain of the MESMAR (v1) regional climate model, showing the extension of the three modeling components (WRF in the atmosphere, NEMO in the ocean, and HD as hydrology model. Filled contours represent the bathymetry and topography over the NEMO and the WRF domains, respectively. The open boundary condition (OBC) shaded area shows the region of application of the NEMO lateral boundary conditions.**




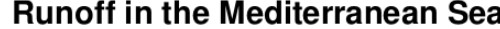

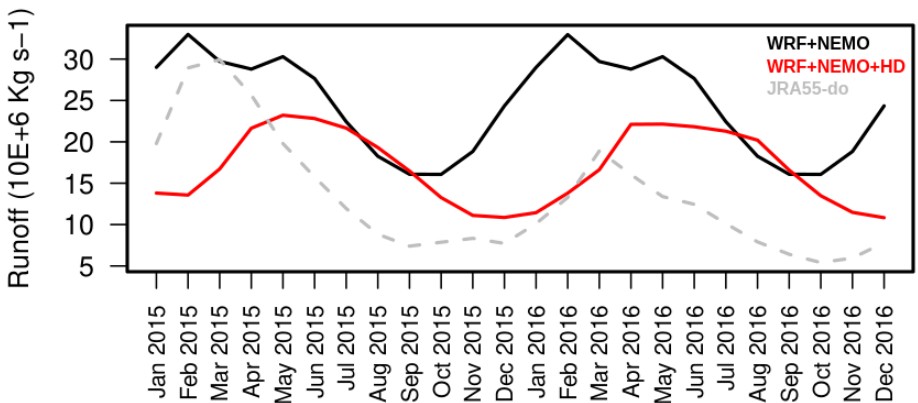

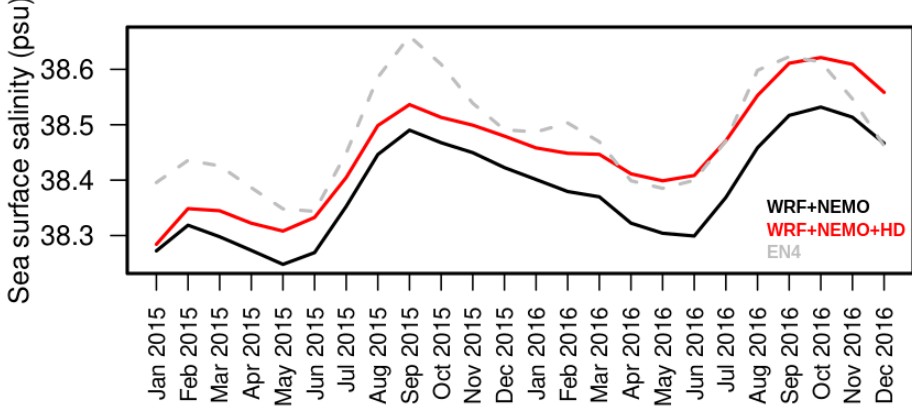

**Figure 2. Monthly means of river discharge into the ocean (top panel) and sea surface salinity (bottom panel) averaged over the Mediterranean Sea, from the experiments with and without the interactive river discharge. Also shows for reference the bias-corrected discharge from the JRA55-do reanalysis and the sea surface salinity from the UKMO EN4 objective analyses.**


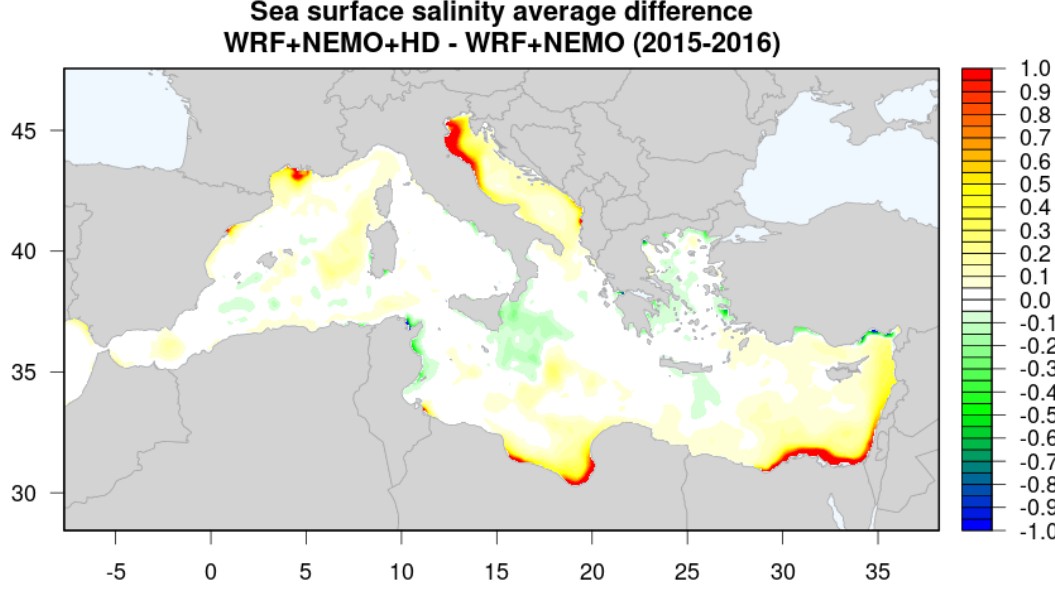


**Figure 3. Average sea surface salinity difference (2015-2016) between the experiments with and without the interactive river discharge, over the Mediterranean Sea. Values are in practical salinity units (psu).**



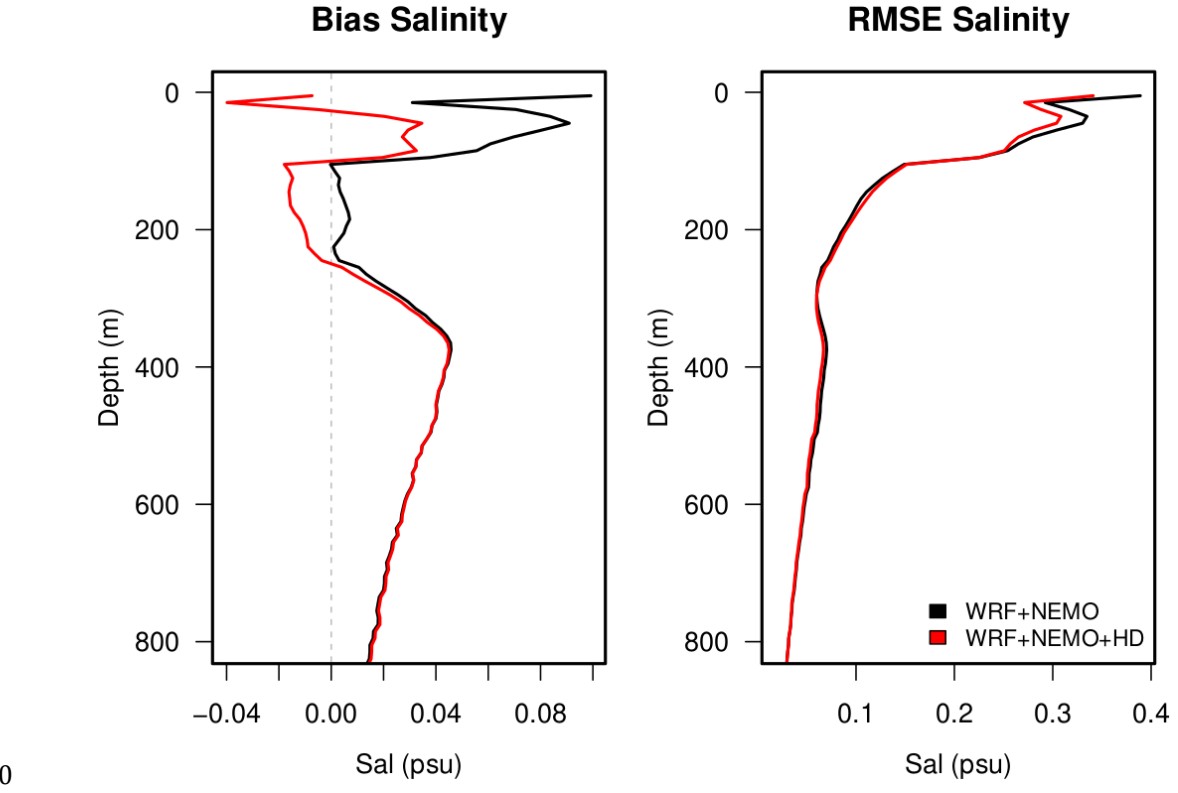


**Figure 4. Profiles of bias and RMSE against observations from Argo floats (EN4 profile dataset) for the experiments with and without the interactive river discharge, over the Mediterranean Sea.**






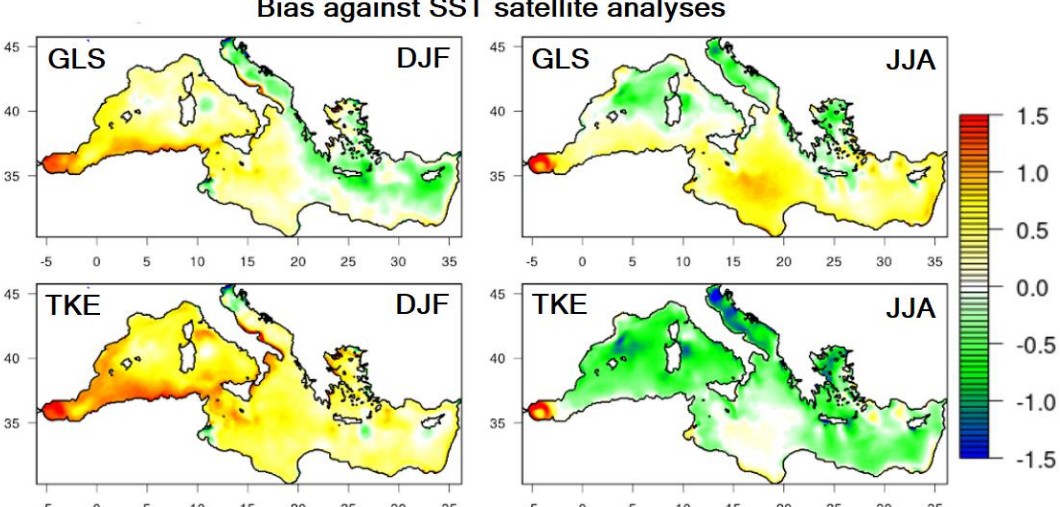

**Figure 5. Winter (DJF) and summer (JJA) biases of sea surface temperature against satellite-based analyses from the Copernicus Marine Service, for the GLS and TKE experiments with different oceanic vertical mixing schemes.**






**Figure 6. As for Figure 4, for the mean, bias, and RMSE profiles against Argo floats, for the GLS and TKE experiments with different oceanic vertical mixing schemes. Mean profiles also report data from the observations (in gray).**






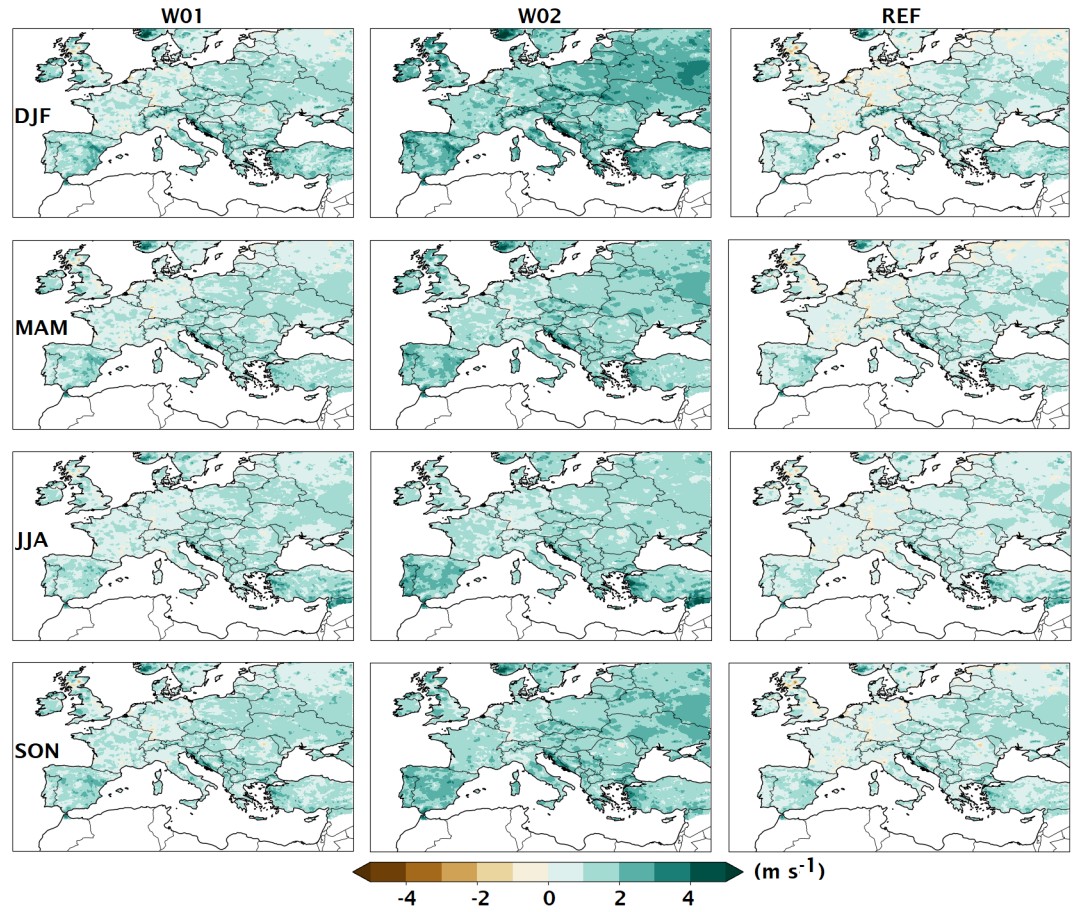

**Figure 7. Differences between the three experiments presented in the text and the E-OBS wind speed (m s⁻¹) for the four seasons DJF, MAM, JJA and SON.**






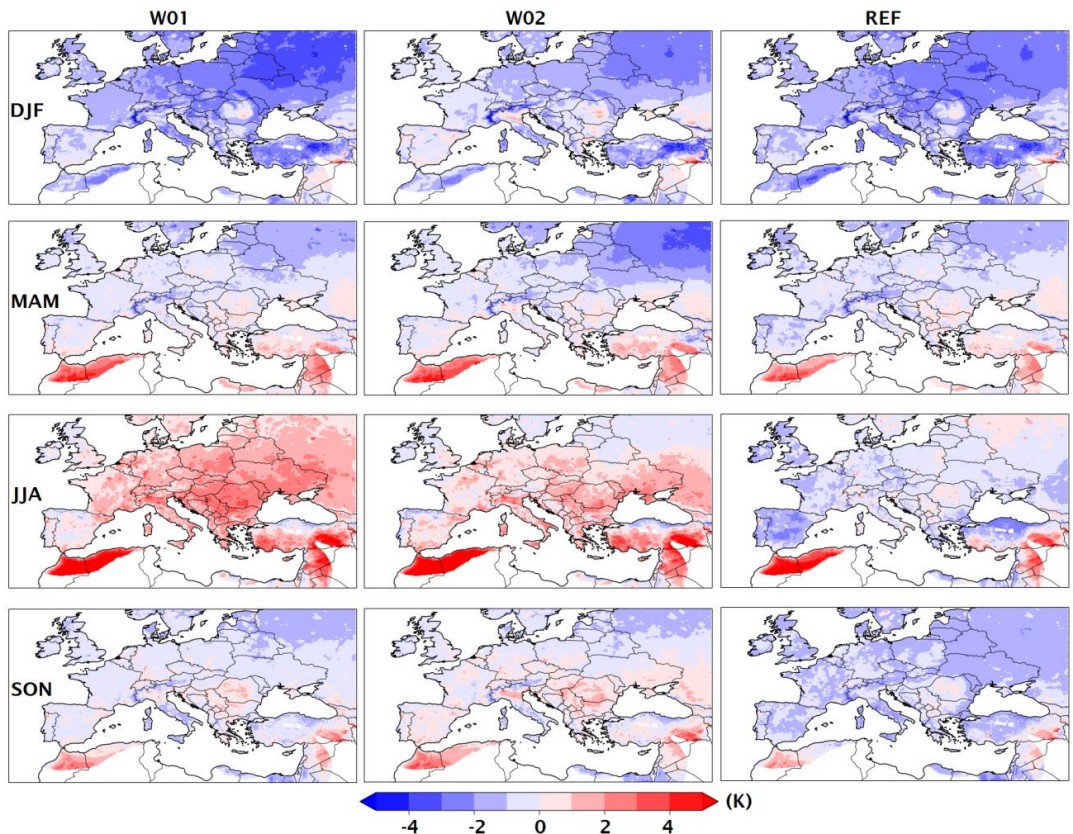

Figure 8. As for Figure 7, for the air temperature at 2 m above ground level (K).






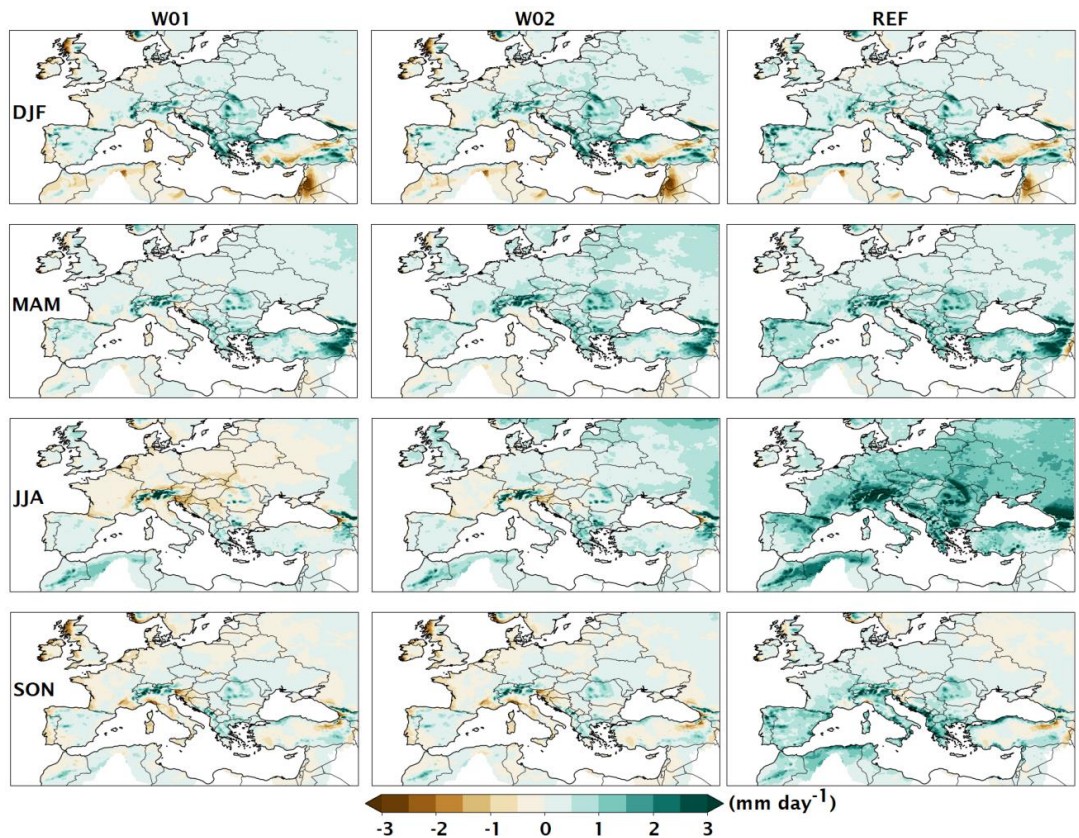

**Figure 9. As for Figure 7, for the total precipitation rate (mm day⁻¹).**




**Figure 10. Mediterranean Sea upper ocean (0-700 m) ocean heat content (OHC, top left panel), incoming heat transport at the Gibraltar Strait (top right panel), and net air-sea heat flux (downward, bottom panel), during the 1993-2020 period, for the MESMAR reference simulation. Also shown for comparison values of OHC from the Copernicus Marine Service Ocean Monitoring Index (OMI), Gibraltar heat transport from the Copernicus Marine Service regional reanalysis, and net air-sea flux from the ECMWF ERA5 reanalysis. The top left panel reports also the OHC timeseries from MESMAR, rectified with the observed long-term OHC trend (red dashed line), while its legend indicates the OHC linear trend (into brackets).**

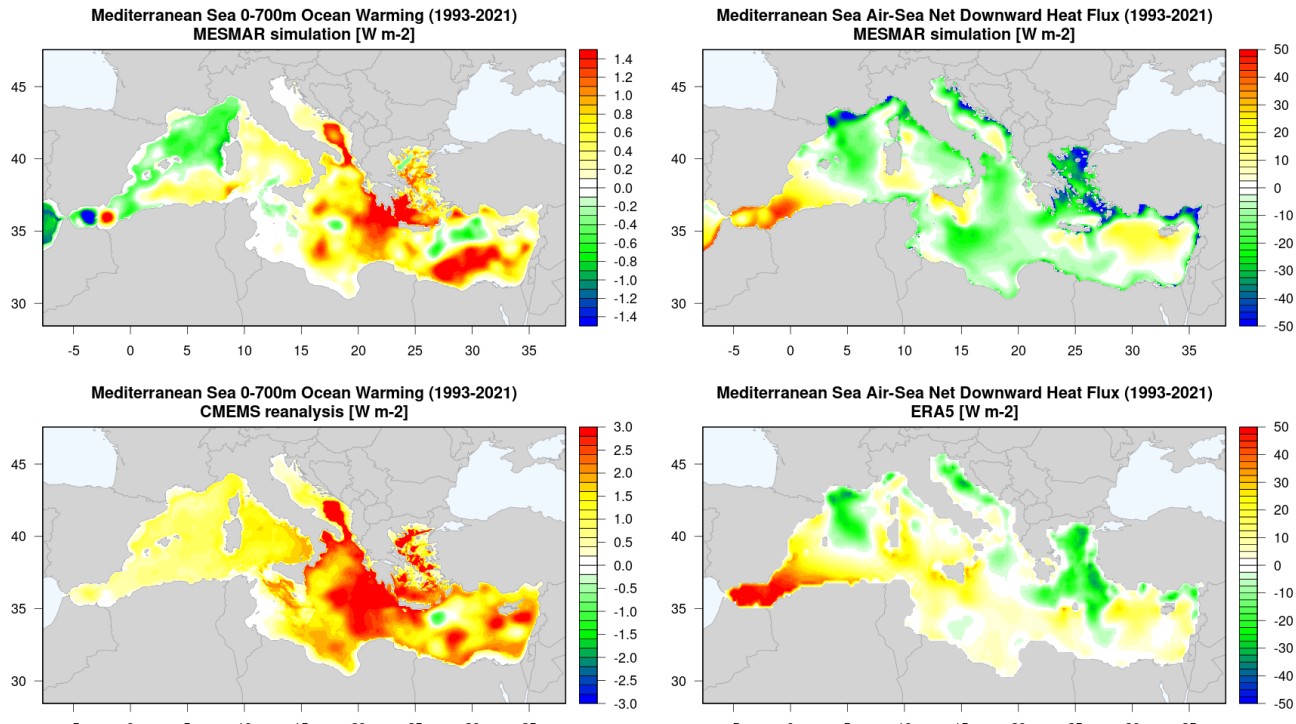

**Figure 11. Upper ocean (0-700m) Ocean warming (OHC linear trend) from the MESMAR reference simulation (top left panel) and the**
**Copernicus Marine Service regional reanalysis (bottom left panel) and long-term mean net air-sea flux from the MESMAR reference simulation (top right panel) and the ECMWF ERA5 reanalysis (bottom right panel).**







**Figure 12. Skill score metrics (bias and RMSE) profiles for the data assimilation experiments calculated for selected atmospheric parameters (air temperature and humidity, wind speed and direction) against radiosonde observations extracted from the RUC NOAA/ESRL archive.**







**Figure 13. As for Figure 12, for the oceanic skill score metrics profiles calculated against Argo float data extracted from the UKMO EN4 profile dataset.**








**Figure 14. SST RMSE differences between AT1OC0 and AT2OC0 (top panel) and between AT2OC0 and AT2OC1 (bottom panel) to show, respectively, the impact of spectral nudging and oceanic data assimilation on the SST RMSE, calculated against the Copernicus Marine Service satellite-based analyses.**






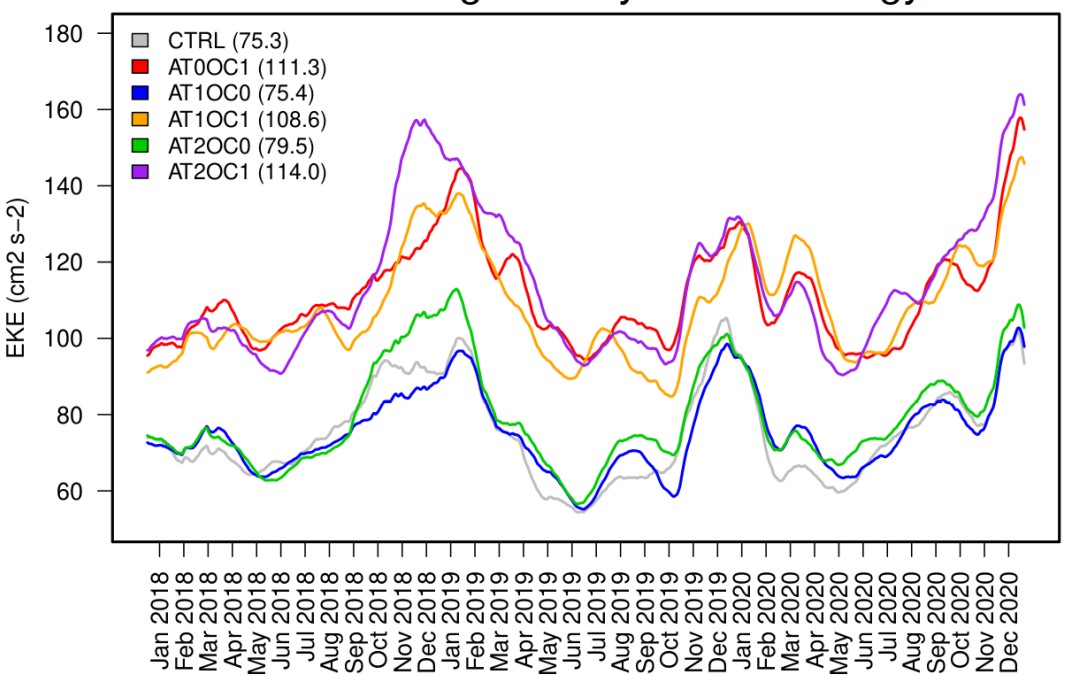

**Figure 15.** Eddy kinetic energy (EKE) over the Mediterranean Sea for the period 2018-2020 and the different data assimilation experiments presented in the text. The EKE is calculated from the sea surface height using geostrophic velocities.




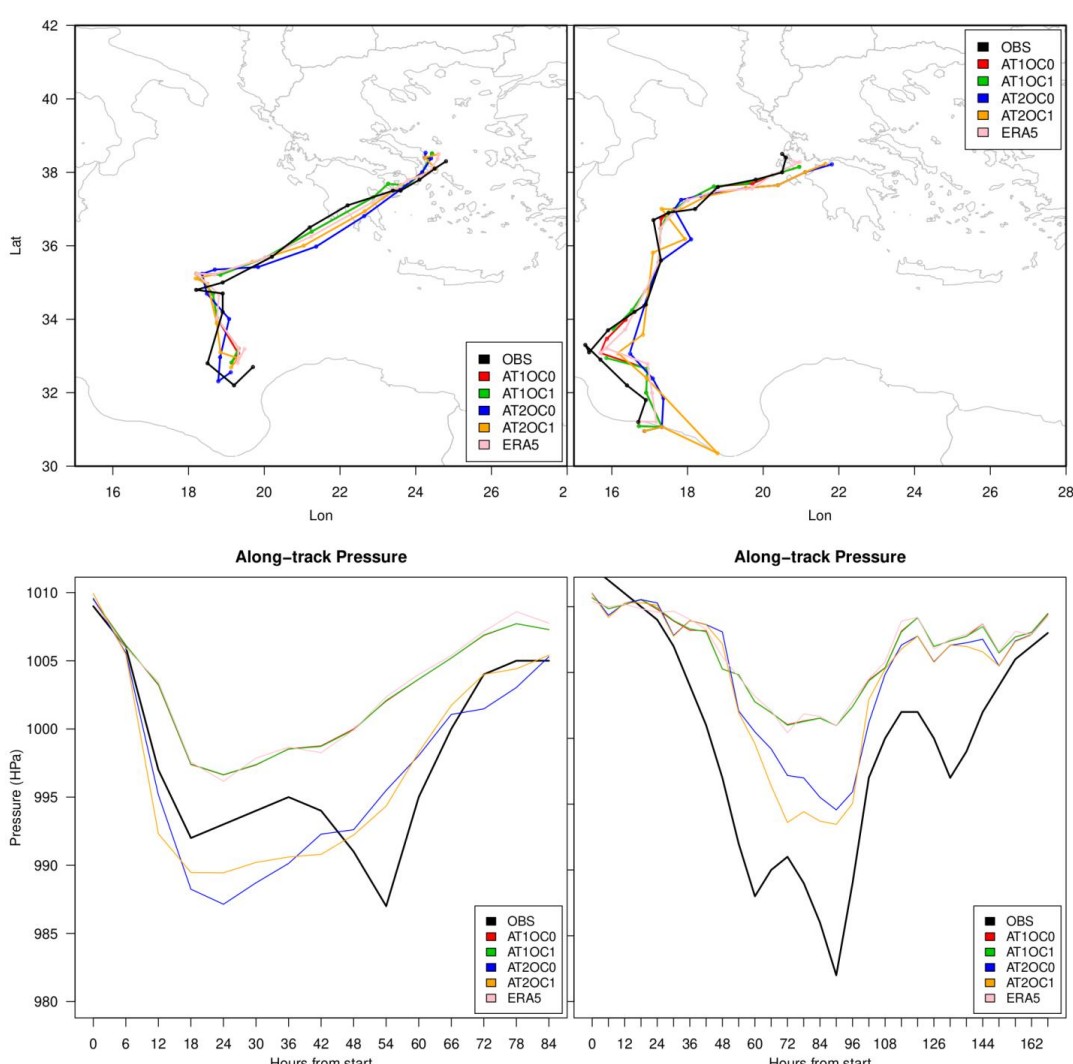

**Figure 16. Medicane tracks (top panels) and along-track sea level pressure (bottom panels) during the two medicane events presented in the text (Zorbas, left panels, and Ianos, right panels). The experiments are run in reanalysis mode (continuous data assimilation), and the ECMWF ERA5 reanalysis is shown for comparison.**

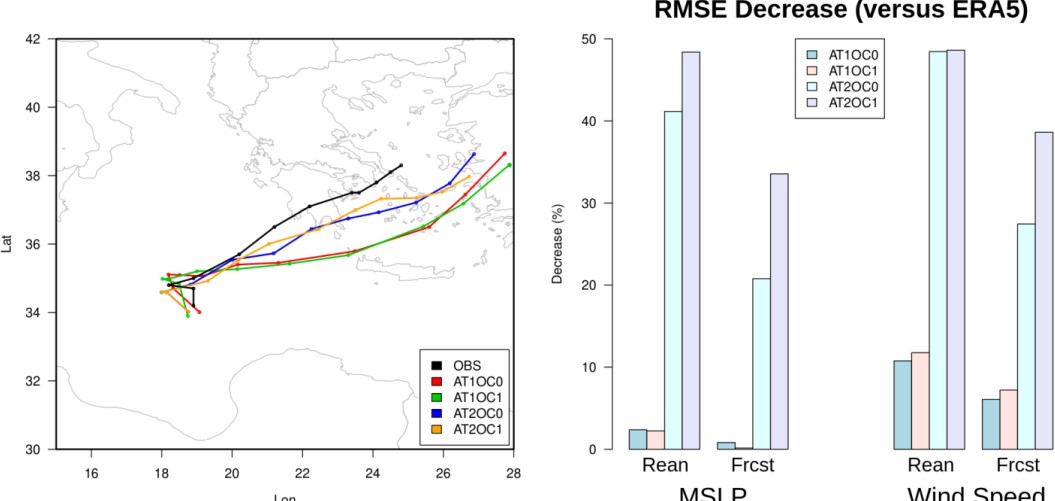

**Figure 17. Left panel: as for Figure 16 (top left panel), but for the forecasts initialized on 28 September with the different data**
**assimilation configurations and run in forecast mode. Right panel: RMSE percent decrease (positive percentage means improvement)**
**compared to the corresponding ECMWF ERA5 reanalysis and forecast, for mean sea level pressure (MSLP) and wind speed.**