# Peer review of "MESMAR v1: A new regional coupled climate model for downscaling, predictability, and data assimilation studies in the Mediterranean region"

_Geoscientific Model Development, 2023_

## Referee Comment (RC2)

**Summary**

This study utilizes a newly developed regional coupled model (atmosphere-ocean-hydrology) to investigate the basic air-sea interactions in the Mediterranean region. Especially, different configurations for each module, such as turbulent parameterization, microphysics and so on, have been tested and the best configuration has been suggested. Further, assimilation experiments were conducted and results related to extreme events were presented.

This manuscript is generally well-written and well-structured. The result of this work is important to the regional climate research and prediction community. I therefore recommend Minor revision after a few modifications.

**Recommendation: Minor revision**

**Minor Points:**

1) Sections 2.1 and 2.2: Have you considered the possibility of using non-hydrostatic core under the current resolution setting? Especially for the cyclone study, the non-hydrostatic process is important for the atmosphere.

2) The resolution for the ocean reanalysis is too coarse. Have you

tried higher-resolution products? Such as The GLORYS12V1 product (the CMEMS global ocean eddy-resolving, 1/12° horizontal resolution, 50 vertical levels).

3) Line 195: It is better to introduce all the observation and reanalysis products in the Data Section.

4) Line 227: s-1 should be $s^{-1}$. Please check similar mistakes in other parts of this manuscript.

5) Line 228-229: A little explanation for the wetter problem?

6) Line 299: What is the nudging time scale for the SST and SSS, respectively?

---

## Author Comment (AC1)

**Revision of "MESMAR v1: A new regional coupled climate model for downscaling, predictability, and data assimilation studies in the Mediterranean region" by Storto et al.(2023)**

**REVIEWER 1**

We thank the reviewer for the encouraging comments and for the suggestions to improve the quality of the manuscript. Below, we provide a point-by-point reply (reviewer in bold, our answer in light font). All coauthors concur with the proposed changes. We refer to the revised version of the manuscript in answering the questions.

Abstract

**Line 17: I would report the horizontal resolution of the models described rather than using the adjective "moderate" or "coarser"**
Corrected, thanks

**Line 21: Are you planning to use the model to produce climate projections? Please specify: The Authors use several times the word "predictions" but never "projections"**
We have no plan to use this system for downscaling projections, so we prefer not to add any reference to it. We added "long-range" to predictions, so that readers can understand we don't mention projections.

**Line 23: "Intense Mediterranean cyclones "rather than "intense mid-latitude cyclones"**
Corrected

**Introduction**

**Line 38: Usually RCM refers to Regional climate models and thus atmosphere not to coupled systems (RCSMs for example, Reale et al., 2022). Please correct that in the text to avoid confusion**
We add a sentence to avoid confusion, thanks. We now mention that RCMs are usually intended to be atmospheric models with physics suites targeted to long-term studies, but recently tend to include an interactive ocean model component.

**Line 38: Please update Giorgi, (1990) that is a bit old as reference**
Updated, thanks

**Line 51: "Coupled…unexplored"..see Sevault et al., 2014. The coupled CNRM model uses the spectral nudging.**
What we mean by "coupled data assimilation" is the simultaneous use of both ocean and atmosphere observations. The Sevault et al. setup includes spectral nudging in the atmosphere, only, but not in the ocean, so we do not consider it as an example of regional coupled data assimilation. These, to our knowledge, are very few, and anyway missing in any Mediterranean Sea system. We prefer to keep the sentence as it is.

**The Authors use several times the word "predictions" but never "projections". Are you planning to use your model also to produce projections? Please specify**
Please see the answer above. We prefer not to specify what we do not want to do (no plan to use it for projections, but you never know).

**Line 72: Be cautious since hurricanes are (from some points of view) very different from medicanes. See Flaounas et al., 2022. Please reformulate this sentence**
Modified, thanks.

**In the introduction it is missing a clear description of the present state concerning the coupled models in the MED region and at which extent the new modeling system described by the Authors is different or eventually represent an advancement with respect the preexisting modeling tools.**
We have added a sentence on the specificities and advancement of our system. We also expanded the discussion about this important point in the Summary and Discussion section, also to respond to

the last comment of the reviewer. In the Intro, we added a sentence about the uniqueness of our modeling system.

**2.Earth system model configuration**

**Line 94: What do you mean with stationary geophysical fields? Is the topography a geophysical field? Please explain**
Yes, plus many more. We use now the term "Geographical Static Data" as given by WRF (https://www2.mmm.ucar.edu/wrf/users/download/get_sources_wps_geog.html) and detail what they are.

**Line 101: Do you refer to the width of the sponge layer? Please specify**
Yes, this is now specified.

**Line 110-115: As far as I understand you are using shortwave radiation as forcing that attenuates along the water column according to a water attenuation coefficient plus the chl-a concentration: since you are using the satellite chl-a (first 10 m as far as I remember) how do you quantify the attenuation effect led by chl-a below 10 m?**
This is detailed in the paper cited: 2d chlorophyll fields are combined with a depth-dependent function to provide 3D attenuation factors. We modified the text to make this more clear.

**Line 117-120: Are you using an open boundary in the Atlantic? Are you applying also the sea surface height at boundary?**
Yes, the description refers to an open boundary. The Flather scheme corrects simultaneously the inner barotropic velocities and ssh according to the external gravity waves. We modified the sentence accordingly.

**Line 126-128: Is the Nile missing from the numerical settings? It is not mentioned in the text.**
Not sure to understand the question: the HD model implements, by construction, a European setup that includes all river basins for the European seas (see fig1), including the Nile. We only mention the Dardanelles because they are a strait and not a river, so HD cannot resolve the freshwater incoming from there

**Line 138: Is "25" resolution dependent? Please explain**
This has been tested only in this configuration, so we cannot answer if the choice of 5x5=25 gridpoints depends on the resolution of MESMAR or not. So, we prefer to keep the sentence unchanged (we anyway corrected 25 to 5x5 to make it clear that it is a rectangle centered on a gridpoint).

**Line 143: Do you mean that all the model components exchange field every 30 min? please specify**
Yes. We changed it to "for all exchanged fields" to make it more clear.

**Line 148-150: Not very clear. Please reformulate. As far as I understand WRF transfers surface and subsurface runoff to HD that is remapped on the ocean grid and passed to NEMO after HD runs. Does HD pass only the river discharge to NEMO? Do you need to locate the river mouths on the ocean model domain?**
HD implements the remapping onto NEMO directly (aware of the NEMO mask, of course), so it passes to NEMO the river runoff directly on the actual NEMO grid. We modified the sentence to make clear HD remaps the runoff onto the NEMO river mouths (OASIS does not perform any interpolation in this case).

**Line 152-157: Why do not use CMEMS MYOCEAN reanalysis specifically tuned for the MED sea instead of GLORYS? Moreover, there are no information about the spin up of your model for pot temperature and salinity. Did you perform the spin up at least for the long run? I would also include additional information concerning the computational performances of the model (number of cores, computational times etc.). It would be useful having a table summarizing the main settings of the model at least in the most important experiments.**

GLORYS12 is preferred to the CMEMS MED reanalysis because the resolution is closer to our model implementation (~1/12 for both systems). We added this and other information about the computational resources in the text.

**3.Sensitivity experiments**

**3.1 Impact of the interactive river discharge**

**I do understand the idea of the Authors to show the importance of river online on the simulate salinity. However, I think that 2 years of runs are too small to assess the importance of river inflow on salinity (in particular along the water column where the signal need sometimes to penetrate) in absence of information about the spin up. I would change Fig.3 and Fig 4 adding the comparison with observations (instead of the comparison between the two configurations) as You did in Figure 6. Moreover, why did you use EN4 instead CMEMS or JRA55 instead of Ludwig et al., 2009.**

Sorry, but the suggestions are not clear: Figure 4 already shows the comparison with observations as suggested by the Reviewer. Not clear what is meant for CMEMS here (reanalyses, observations, objective analyses?), in any case, EN4 is a state-of-the-art profile and objective analysis product that we use for validation (in Fig 3 as SSS analyses, in Fig 4 as raw observation data). There is no reason why we should use CMEMS (and why not the NOAA dataset then? Or the JMA one?).

We clarified this point in the text. JRA55-do is a daily reprocessed dataset, while Ludwig 2009 is a climatology, so it is more appropriate as verifying dataset for our 2-year experiment. We modified the text to stress that the experimental period is short, so the results are only indicative of the potential of HD coupling.

**3.2 Nemo vertical physics**

**There are no information about the length of tests. Moreover, there is no quantification of biases (smaller etc is too generic). Please quantify the biases**

Thanks, we added this information in section 3.2

**3.3 WRF configuration**

**Why have your tests run only for the period 1993-2021 (also in the reference simulation) instead of covering the entire ERA5-ORAS5 period? Please specify**

This is a standard period that includes altimetry, and it is used, for instance, for CMEMS reanalysis dissemination, C3S seasonal prediction forecasts, and more. Of course, the choice is arbitrary but consistent with other datasets. We do not think we need to justify the choice of the experimental period, so we leave the text as it is.

**What do not you investigate also the behavior of wind and precipitation on the ocean domain since their importance for E-P and mixing?**

The impact was found not significant on the ocean bias, we added a comment on this at the end of the section.

**4 Reference simulation**

**Line 235: I would add also Soto-Navarro et al., (2020) and Reale et al., (2022) that addressed this issue.**

**Soto-Navarro, J., Jordá, G., Amores, A., Cabos, W., Somot, S., Sevault, F., et al. (2020). Evolution of Mediterranean Sea water properties under climate change scenarios in the Med-CORDEX ensemble. Clim. Dyn. 54, 2135–2165. doi: 10.1007/s00382-019-05105-4**

**Reale, M., Cossarini, G., Lazzari, P., Lovato, T., Bolzon, G., Masina, S., ... & Salon, S. (2022). Acidification, deoxygenation, and nutrient and biomass declines in a warming Mediterranean Sea. Biogeosciences, 19(17), 4035-4065.**

Added, thank you.

**Being a long term simulation, it would be interested having the analysis of long term timeseries of temperature and salinity with respect to observational datasets to have an idea about the behavior of the thermohaline properties at different depths.**
Thanks for the suggestion. We report below a figure showing the Mean, Bias, and RMSE of temperature in observation space (profile data). We include a comment on this in the text, but we think the figure is not very informative to be included.

[Figure]

*Mean, bias and RMSE of T in observation space for the entire basin*

**It would be important also to assess the source of the error in the total net heat flux. Is it an underestimation of the net shortwave? Latent or sensible heat fluxes? Please think to include this analysis**

This was already stated, at the end of section 4, where we discussed the difference of the individual air-sea heat flux components.

**Figure 11 Why is the area outside Gibraltar colored in the upper panels and not in the bottom panels?**
We have now masked out areas outside Gibraltar in all panels, for consistency between panels, thanks.

**5 Data assimilation**

**I found the data assimilation an interesting new advancement in the coupled model. However, I do not understand the reason for relaxing at the surface when already both atmosphere and ocean assimilate at high frequency data to correct errors in the simulated field. Could you please explain better that?**
As this system is intended also for long runs and reanalyses, SST relaxation provides a temporally consistent way to ingest sea surface data without spurious variations linked to the observational sampling. This is a very standard approach used in most state-of-the-art ocean reanalyses (ORAS5, CGLORS, etc.). We shortly explained this in the text. Note, there is no redundancy of data, as the variational DA assimilates profiles, and the nudging ingests sea surface temperature data (from satellite, referenced to drifter SST data).

**Moreover, I would ask the Authors to better discuss why with fully data assimilation active (and high frequency of assimilation) the improvements in the biases is relatively small. Low quality of input data? Please infer on that.**
We do not agree with the reviewer. The impact of DA is significant. Looking at Figure 13 bottom middle panel, for instance, we see a large impact of DA. Table 3 quantifies the impact. This is reflected in all ocean skill scores when ocean DA is switched on.

**5 Mediterranean hurricanes How do you track and reconstruct the temporal evolution of the systems analyzed in the manuscript? Please infer on that.**
Tracks correspond to the surface pressure minimum. We added this in the main text.

**6 Discussion and conclusions There is a lack of the comparison with respect to previous modeling systems. Why should a potential user use your modeling tool instead of another one? Do you expect that increasing the horizontal resolution should improve the performances of your model or should make it slower?**
Thanks for pointing this issue out. We have expanded the first paragraph of the Summary section to address this point, and the one mentioned above by Reviewer 1 in the Introduction, and pointed out that the resolution increase is intended for short experiments (e.g. mimicking short-range forecasts).

---

## Author Comment (AC2)

**Revision of "MESMAR v1: A new regional coupled climate model for downscaling, predictability, and data assimilation studies in the Mediterranean region" by Storto et al.(2023)**

**REVIEWER 2**

We thank the reviewer for the encouraging comments and for the suggestions to improve the quality of the manuscript. Below, we provide a point-by-point reply (reviewer in bold, our answer in light font). All coauthors concur with the proposed changes. We refer to the revised version of the manuscript in answering the questions.

**Minor Points:**

**1) Sections 2.1 and 2.2: Have you considered the possibility of using non-hydrostatic core under the current resolution setting? Especially for the cyclone study, the non-hydrostatic process is important for the atmosphere.**
Our WRF implementation uses a non-hydrostatic core. We state it clearly in the revised version, as this information was missing. Thanks.

**2) The resolution for the ocean reanalysis is too coarse. Have you tried higher-resolution products? Such as The GLORYS12V1 product (the CMEMS global ocean eddy-resolving, 1/12° horizontal resolution, 50 vertical levels).**
Within preliminary runs, we compared all the CMEMS reanalyses, but we did not find any significant difference in terms of the impact of the lateral BCs, except (as already mentioned in the manuscript) the positive impact of using ORAS5 on the SSH skill scores (due to barotropic transport at the Atlantic boundaries). We do use GLORYS12 as initial conditions, as specified in the paper, as it provides better spatial detail. We modified the text (line 125) to explicitly refer to this.

**3) Line 195: It is better to introduce all the observation and reanalysis products in the Data Section.**
Thanks, we moved the description of the verification dataset at the end of section 2.5.

**4) Line 227: s-1 should be s-1. Please check similar mistakes in other parts of this manuscript.**
Corrected thanks

**5) Line 228-229: A little explanation for the wetter problem?**
It is quite difficult to disentangle the impact of all parametrizations. We tested a large combination of options in a long preliminary series of sensitivity experiments. The interactions between microphysics, radiations, PBL, etc. are not linear and do not allow a robust answer to the Reviewer's question. We now refer to results in the literature to speculate about the reasons.

**6) Line 299: What is the nudging time scale for the SST and SSS, respectively?**
We are not sure to understand the question: the sentence reads "The relaxation time scales are set equal to 15 and 300 days for SST and SSS, respectively" so it already contains the nudging time scale. We have kept the sentence unchanged.

---

## Author Comment (AC3)

**Revision of "MESMAR v1: A new regional coupled climate model for downscaling, predictability, and data assimilation studies in the Mediterranean region" by Storto et al.(2023)**

**REVIEWER 3**

We thank the reviewer for the encouraging comments and for the suggestions to improve the quality of the manuscript. Below, we provide a point-by-point reply (reviewer in bold, our answer in light font). All coauthors concur with the proposed changes. We refer to the revised version of the manuscript in answering the questions.

**Minor Points:**

**In agreement with another reviewer, I would like some more discussion (in the Intro.) on why the system is different from previous regional models, and its specific advantages. There is a brief discussion at lines 379-383, but I would like it more explicit in the Introduction. For example, to what extent do other systems have data assimilation?**
We have added a sentence on the specificities of our system, according also to Reviewer #1 question.

**Line 17. It would help to specify the grid spacing for ocean and atmosphere (not just ocean) using the same units.**
Corrected, thanks, as requested also by Reviewer #1

**The paper has a slight bias towards the oceans (with less discussion of atmosphere processes). This is OK, but lines 39-43 ignore the fact that enhanced resolution may improve representation of atmosphere dynamics, not just ocean. Is one of the aims of your paper to identify when full coupling provides better results? You could make this explicit.**
We reformulate this sentence according to Reviewer 1, pointing out that RCMs were originally only atmospheric models, and when coupled to the ocean a better representation of the air-sea fluxes is generally provided. So, we believe the reviewer's comment is important, but the new version of the paragraph overcomes this point.

**Lines 56-57 and 66-67 both introduce medicanes. I suggest to delete the first example.**
Yes, deleted, thanks

**Line 101. Are the lateral bcs applied as some kind of nudging?**
Exactly, added

**Line 108 "partial bottom steps"**
Corrected.

**Lines 114-116. Is this the same as the GOTM (https://gotm.net/portfolio/) ?**
Yes, however, we are not fully sure that the technical implementation of GLS in NEMO is exactly as in GOTM, so we prefer not to mention GOTM.

**Line 177. 2 years seems a bit short for this comparison.**
We agree and have made explicit in the manuscript that the results are partial.

**Line 183. For Fig. 4, add a map of model minus EN4 surface salinity (probably annual mean).**
Thanks for the suggestion, we added a panel accordingly (in Figure 3).

**Line 193 introduces the TKE scheme in the wrong place. It should be described more fully in section 2.2.**
Done.

**Lines 205-210. Fig. 5 shows a strong seasonal dependence of bias, but Fig. 6 includes all seasons. Consider making the equivalent of Fig. 6 for winter and summer.**
Thanks for the suggestion. We show here the figure and we comment it in the text, but the number of figures is already very large so we prefer not to add more figures.

*As for Figure 6, but for the RMSE of Temperature in DJF and JJA.*

**Lines 270-275. Does the net air-sea heat flux difference between MESMAR and reanalysis relate to SST difference and turbulent fluxes, or to radiative fluxes?**
We detail this at the end of the section: roughly 50% is due to turbulent fluxes (overestimated in MESMAR, ie greater heat loss) and 50% to underestimated incoming solar radiation.

**Line 288. "climatological anomly"-> "anomalies from climatology"?**
Corrected

**Line 305. More definition of full-field nudging - I assume this means no length-scale filter, and similar timescales?**
Added, thanks

**Line 320. "SST skill scores"    - SST is nudged to analysis on 15-day timescales, so i don't know how to interpret "SST skill"!**

The effect of SST scores is discussed when commenting on Figure 14. Maybe it was not clear in the text: the SST nudging is switched on only in the "OC1" experiments, ie only when also variational assimilation of profiles is switched on. We clarified this in the text and table.

**Section 5.2. I assume that the verification data was not assimilated. (Please state this explicitly.)**
We added this in section 2.5

**Fig. 14. As above, the system is nudged to SST, so how to interpret MESMAR minus satellite SST?**
Not really, the experiments OC0 have no data assimilation so the comparison in the top panel has no nudging. In the bottom panel, instead, we want to evaluate the combined effect of all ocean observations (nudging and variational assimilation). We have clarified this in the text.

**Line 357. CTL has Medicanes due to bcs?**
They are not shown because the track is very far and the intensity is much smaller than in the other experiments. Please note, boundary conditions are far from the tracks, and with no data assimilation nor cycled initialization, CTL can represent medicanes only "statistically".

Lines 362, 375 quote percentage improvements for ocean assimilation, how do they compare to **atmosphere assimilation? I understand that we expect ocean assimilation to have a lesser effect than atmosphere assimilation for atmospheric weather systems, but it might be useful to compare.**
AT0OC1 has indeed a similar representation as CTL. With no atmospheric data assimilation, there is no way to correctly capture the location of the medicane. This is already stated in the text; the question we wanted to investigate is whether the ocean DA adds value on top of the atmospheric DA. This is the case at least for the medicane intensity. We better clarified this in the text.

**Fig. 11, consider adding two more panels of the differences: model minus reanalysis.**
Added and commented on the text. Thanks

**Fig. 12 - I think a lot of lines overly each other, e.g. BIAS of humidity, perhaps mention this in the text. i.e. the lack of sensitivity to changing configuration.**
We now mention it in line 347.